# Druggable redox pathways against *Mycobacterium abscessus* in cystic fibrosis patient-derived airway organoids

Stephen Adonai Leon-Icaza[1], Salimata Bagayoko[1], Romain Vergé[1], Nino Iakobachvili[2], Chloé Ferrand[1], Talip Aydogan[3], Célia Bernard[1], Angelique Sanchez Dafun[1], Marlène Murris-Espin[4,5], Julien Mazières[4], Pierre Jean Bordignon[1], Serge Mazères[1], Pascale Bernes-Lasserre[6], Victoria Ramé[6], Jean-Michel Lagarde[6], Julien Marcoux[1], Marie-Pierre Bousquet[1], Christian Chalut[1], Christophe Guilhot[1], Hans Clevers[7], Peter J. Peters[2], Virginie Molle[3], Geanncarlo Lugo-Villarino[1], Kaymeuang Cam[1], Laurence Berry[3], Etienne Meunier[1], Céline Cougoule[1] *

1 Institut de Pharmacologie et de Biologie Structurale (IPBS), Université de Toulouse, CNRS, Université Toulouse III–Paul Sabatier (UPS), Toulouse, France, 2 M4i Nanoscopy Division, Maastricht University, Maastricht, Netherlands, 3 Laboratory of Pathogen Host Interactions (LPHI), Université Montpellier, CNRS, Montpellier, France, 4 Service de Pneumologie, Hôpital Larrey, CHU de Toulouse, Toulouse, France, 5 Centre de ressource et de compétence pour la mucoviscidose de l'adulte (CRCM adulte), CHU de Toulouse, Toulouse, France, 6 Imactiv-3D SAS, 1 Place Pierre POTIER, Toulouse, France, 7 Oncode Institute, Hubrecht Institute, Royal Netherlands Academy of Arts and Sciences and University Medical Center, Utrecht, Netherlands

* Celine.Cougoule@ipbs.fr

**Data Availability Statement:** The authors confirm that all data underlying the findings are fully available without restriction. All relevant data are

## Abstract

*Mycobacterium abscessus* (Mabs) drives life-shortening mortality in cystic fibrosis (CF) patients, primarily because of its resistance to chemotherapeutic agents. To date, our knowledge on the host and bacterial determinants driving Mabs pathology in CF patient lung remains rudimentary. Here, we used human airway organoids (AOs) microinjected with smooth (S) or rough (R-)Mabs to evaluate bacteria fitness, host responses to infection, and new treatment efficacy. We show that S Mabs formed biofilm, and R Mabs formed cord serpentines and displayed a higher virulence. While Mabs infection triggers enhanced oxidative stress, pharmacological activation of antioxidant pathways resulted in better control of Mabs growth and reduced virulence. Genetic and pharmacological inhibition of the CFTR is associated with better growth and higher virulence of S and R Mabs. Finally, pharmacological activation of antioxidant pathways inhibited Mabs growth, at least in part through the quinone oxidoreductase NQO1, and improved efficacy in combination with cefoxitin, a first line antibiotic. In conclusion, we have established AOs as a suitable human system to decipher mechanisms of CF-driven respiratory infection by Mabs and propose boosting of the NRF2-NQO1 axis as a potential host-directed strategy to improve Mabs infection control.

within the paper and its Supporting Information files.

**Funding:** This project has been funded by grants from: "Vaincre La Mucoviscidose" and "Grégory Lemarchal" foundations (N° RF20210502852/1/1/48) to CCo, including the PhD fellowship for SALI; the "Fondation pour la Recherche Médicale" ("Amorcage Jeunes Equipes", AJE20151034460), the Solvay Solidarity Fund and CNRS Foundation, the CNRS ATIP avenir program, and ERC StG (INFLAME 804249) to EM. SB was funded by "Fondation pour la Recherche Médicale" PhD fellowship (FDT202106012794). This work was also supported by grants from CNRS (IEA 300134) to CCo, Campus France PHC Van Gogh (40577ZE) to GL-V, ZonMW 3R's (114021005) and the LINK program from the Province of Limburg, the Netherlands to PJP, the Nuffic Van Gogh Programme (VGP.17/10) to NI, and by the ANR JCJC ProteasoRegMS to ASD. TA, VM and LB were funded by "La Région Languedoc-Roussillon" (N° DRTE/RSS - ESR_R&S_DF-000061-2018-003268). The funders had no role in study design, data collection and analysis, decision to publish, or preparation of the manuscript.

**Competing interests:** Julien Mazières reports grants or contracts from Astra Zeneca, Roche and Pierre Fabre; and payment or honoraria for board and expertise (personal and institution) from Merck, Astra Zeneca, BMS, MSD, Roche, Novartis, Daiichi, and Pfizer; outside the submitted work. Hans Clevers reports invention on patents related to organoid research. His full disclosure: www.uu.nl/staff/JCClevers/Additional function. The other authors have declared no competing interests.

## Author summary

Pulmonary infection by Non Tuberculosis Mycobacteria is a rising concern for patients with cystic fibrosis (CF), especially Mycobacterium abscessus (Mabs). Mabs exists as two morphotypes. CF patients are generally infected by the S morphotype present in the environment, which can switch to the R morphotype displaying higher virulence. Due to its resistance to antibiotics, treatments againt Mabs often fail, calling for complementary therapeutical strategies. Here we adapted the human airway organoid technology to model Mabs infection in the context of CF, decipher mechanisms of host-pathogen interaction that can be pharmacologically targeted to improve infection control. We found that Mabs R induces higher host oxidative stress and cell death, hallmarks of its virulence, which are enhanced in the CF context. Boosting the host oxidative pathway using antioxidants improves infection control by a frontline antibiotic. Our study provides CF patient-derived airway organoids as a relevant human-based, animal-free system for CF-driven Mabs infection and evaluation of innovative therapeutic strategies.

## Introduction

Cystic Fibrosis (CF) is a monogenic disease due to mutations in the CF transmembrane conductance regulator (CFTR) gene [1], which regulates ion transport, that impair lung mucociliary clearance and result in pathological triad hallmarks of CF, i.e., chronic airway mucus build-up, sustained inflammation, and microbe trapping leading to parenchyma epithelial cell destruction. The major reason CF patients succumbing to this disease is respiratory failure resulting from chronic lung infection [2].

CF Patients have a greater risk of infection by Non-Tuberculous Mycobacteria (NTM), mainly by the drug-resistant NTM *Mycobacterium abscessus* (Mabs) [3–5]. Mabs display two distinct morphotypes based on the presence or absence of glycopeptidolipids (GPL) in their cell wall [6]. The smooth (S) GPL-expressing variant forms biofilm and is associated with environmental isolates. The Rough (R) variant does not express GPL, forms cording and induces more aggressive and invasive pulmonary disease, particularly in CF patients [6–8]. Mabs colonization of the CF patient airway is initiated by the infection with the S variant that, over time, switches to the R morphotype by losing or down-regulating surface GPL [9–11]. Although animal models like immunocompromised mice [12,13], zebrafish [14–16] and *Xenopus laevis* [17] contributed to a significant advance in the understanding of Mabs infection [18], their tissue architecture and cell composition are different from that of humans and do not recapitulate the hallmarks of CF [19–21]. Models with anatomical and functional relevance to the human airway and displaying natural CFTR gene mutations would complement those *in vivo* models.

Human airway organoids (AOs), derived from adult stem cells present in lung tissues [22], are self-organized 3D structures and share important characteristics with adult bronchiolar part of the human lung [22,23]. Of particular interest, organoids derived from CF patients constitute a unique system to model natural CFTR mutations and the resulting epithelium dysfunctions such as exacerbated mucus secretion, thus recapitulating critical aspects of CF in human that are not achievable with other cellular or animal models [22,24,25]. AOs have also been adapted for modelling infectious diseases with bacteria, such as *Pseudomonas aeruginosa* [26], with viruses, such as RSV [22] and SARS-CoV-2 [27–29], and with parasites [30]. We previously showed that *M. abscessus* thrives in AOs [31], demonstrating that AOs constitute a suitable human system to model mycobacteria infection.

Here, we hypothesized that AOs could model Mabs variant virulence and how the CF lung context influence Mabs infection. In this study, we therefore assessed Mabs variant infectivity in AOs, and the influence of CFTR dysfunction using CF patient-derived AOs. We report that both Mabs S and R infect and replicate within AOs and display their specific extracellular features, especially biofilm and cording, respectively. Moreover, enhanced reactive oxygen species (ROS) production during Mabs infection and the CF context favours Mabs growth, which is reversed by antioxidants that improved antibiotic efficacy.

## Results

### Human airway organoids support S- and R-Mabs replication and phenotype

We microinjected bronchial airway organoids with S- and R- Mabs variants (380+94 CFU and 298+44 CFU respectively, mean + SEM, P>0.99, Fig 1A) as previously described [31]. We first quantified bacterial load in AOs overtime (S1A Fig), and showed that both Mabs S and R propagated over 12 days. Based on these data, we then performed experiments at 4 days post-infection, corresponding to bacteria exponential growth phase. We observed a low but significant difference between Mabs S and R, with Mabs S growing better compared to the R morphotype at 4 days post-infection (Fig 1A). We also showed that other Mabs subspecies also replicated in AOs, with *M. abscessus subspecie bolletii* displaying a slightly higher growth than Mabs S or *M. abscessus subspecie massiliense* (S1B Fig), while their growth *in vitro* was similar (S1C Fig). When analysed microscopically, we showed that Mabs S and R mainly resided in the lumen of AOs and we did not detect obvious alteration of the architecture of Mabs-infected AOs compared to mock injected AOs (Fig 1B). Interestingly, by light sheet imaging, we observed that S bacteria formed aggregates in the lumen of the organoids, whereas the R variant formed serpentine cords characterized by the parallel arrangement of the bacteria along their long axis (Figs 1C and S1D–S1I and S1 and S2 Movies), structures observed both *in vitro* and *in vivo* [32]. To further investigate Mabs behaviour, S- and R-Mabs-infected AOs were analysed by SEM and then TEM (Fig 1D). As previously described [22], the organoid epithelium is composed of basal, ciliated and goblet cells (Fig 1D, 1st row). Mabs S bacilli formed chaotically scattered aggregates in the organoid lumen (Fig 1D, 2nd row). We found Mabs S localizing in close contact with the apical side of the epithelial cells (Fig 1D, 4), particularly in the presence of cilia (Fig 1D, 5). The same samples observed by TEM revealed that S bacteria in the lumen were surrounded by what could be an extracellular polymeric substance [33] (Figs 1D, panel 6–6' and S2A), suggesting that Mabs S variant might form biofilm in the organoid lumen. We convincingly identified Mabs R forming serpentine cords, characterized by parallel and aligned bacteria, in the organoid lumen (Figs 1D, panels 7–9 and S2B). Importantly, electron microscopy confirmed no significant internalization of Mabs by epithelial cells [31]. As formation of cords constitutes a mycobacterial virulence trait [11,34,35], we next quantified Mabs cording over time of AO infection. We never observed formation of cords during Mabs S infection, but quantified an increase in the number of organoids containing cords upon Mabs R infection over 4 days of infection (Fig 1E). Finally, we evaluated the virulence of S- and R-Mabs by assessing epithelial cell damage. As shown in Fig 1F and 1G, AOs infected with the R variant exhibited enhanced cell death compared to those infected with the S variant likely reflecting the higher virulence of the cord-forming R morphotype.

Altogether, our results showed that S and R Mabs variants thrived in AOs and displayed respective features observed in *in vivo* models and in the lung of CF patients.

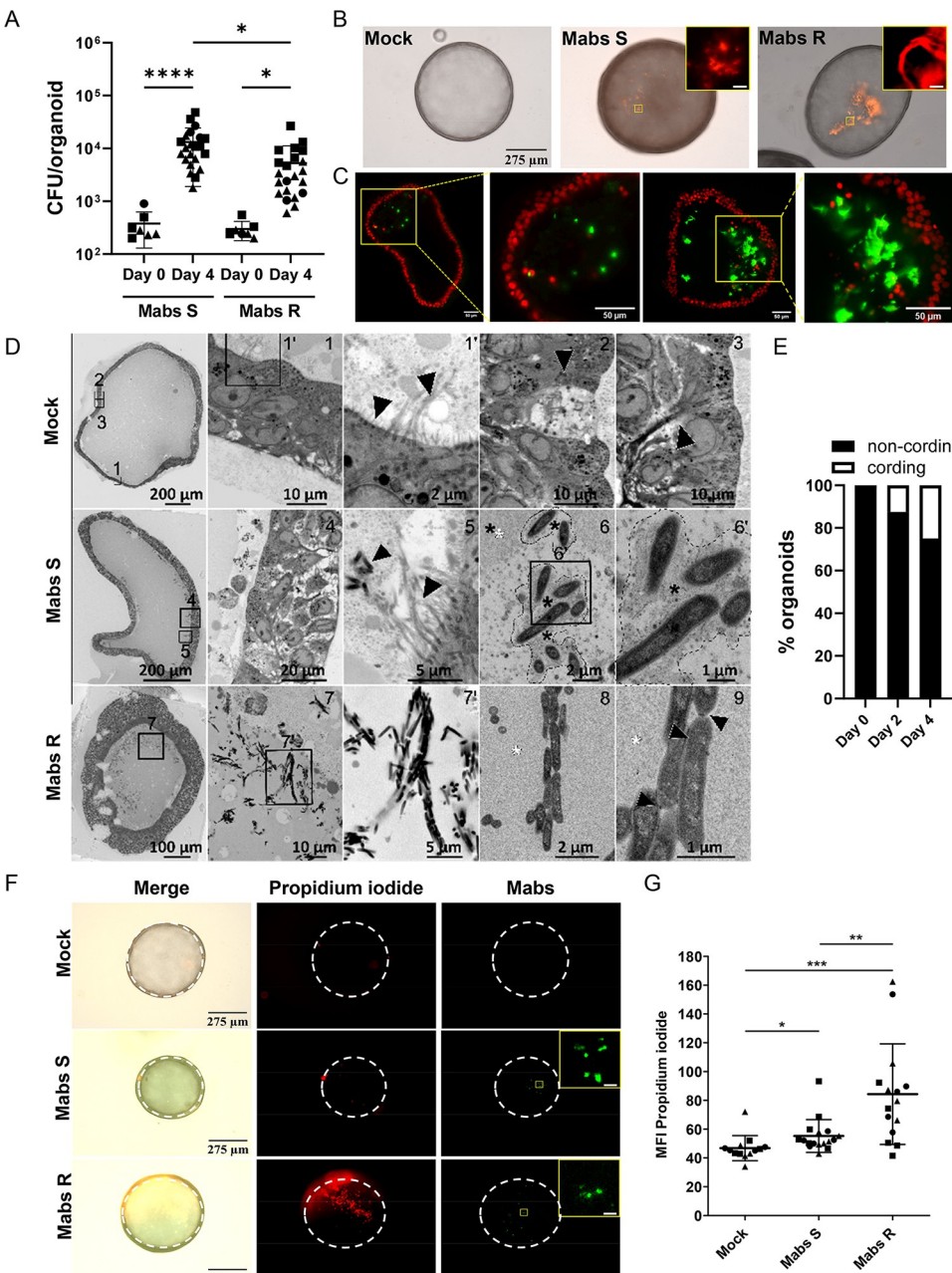

**Fig 1. Mabs infection in airway organoids.** (A) Mabs S (Day 0 n = 7; Day 4 n = 22) and R (Day 0 n = 7; Day 4 n = 22) growth in healthy AOs (H-AO). (B) Representative images of a mock (PBS) infected AO or AOs infected with tdTomato-expressing Mabs S or R. (C) Light-sheet fluorescence microscopy of a XY plane at the z = 120μm (left two images) or z = 80μm (right two images) positions of an AO infected with Mabs S or R, respectively; Zoom-in image of the yellow square zone. (D) Electron micrographs obtained with a FEI Quanta200 scanning electron microscope set up in back-scattered mode. Resin blocks were sectioned and imaged at different magnifications showing normal AO organization and the different cell types typical of lung epithelium (top row, panel 1': left arrow indicates microvilli, right arrow indicates cilia; panel 2: arrow indicates a goblet cell; panel 3: arrow indicates a club cell), the biofilm formed by Mabs S on the luminal face of the epithelial cells (middle row, panel 5: left arrow indicates bacteria; right arrow indicates cilia; panel 6 & 6': asterisks indicate the extracellular polymeric substance surrounding Mabs S), and the bacterial aggregates typical of the cording in the lumen of Mabs R infected AOs (bottom row, panel 9: arrows indicate electron dense deposit lining Mabs R making cords). Targeted ultrathin sections were made and observed by transmission electron microscopy (images 5 and 8). (E) Mean percentage of organoids containing Mabs R cords after 0 (n = 8), 2 (n = 8), or 4 (n = 8) days of infection. (F, G) Representative images (F) and Mean Fluorescence Intensity (MFI) quantification (G) of propidium iodide incorporation (50 μg ml⁻¹) in mock infected AOs (n = 13) or AOs

infected with Wasabi-expressing Mabs S (n = 17) or R (n = 15) for 4 days. The dotted lines delimit the organoids circumference. Except otherwise stated, graphs represent means ± SD from at least two independent experiments, indicated by different symbols. Each dot represents one organoid. *P<0.05; **P<0.01; ***P<0.001 by Mann-Whitney test.

## ROS production contributes to *Mycobacterium abscessus* growth

Next, we compared the host epithelial cell response to Mabs S and R infection by measuring the expression of genes related to inflammatory cytokines, antimicrobial peptides (AMPs), mucins and redox homeostasis in infected airway organoids. As previously described [31], we confirmed a tendency towards a decrease in pro-inflammatory cytokine expression, except for CXCL10 for which expression is significantly induced by both variants (S3A Fig). We also confirmed the modulation of AMPs expression, with a significant inhibition of lactoferrin expression (S3B Fig). Regarding mucin expression, we showed that infection by both S and R Mabs inhibited MUC5B and MUC4 expression (S3C Fig). As ROS production is an important antimicrobial process and is enhanced in the CF context [36,37], we finally evaluated the expression of genes related to the production and detoxification of ROS. While Mabs S did not significantly modulate the expression of NOX1 and DUOX1 oxidases, we measured a significant increase in DUOX1 expression upon Mabs R infection, suggesting enhanced ROS production by this variant (Fig 2A). Moreover, we observed a significant increase in the expression of the transcription factor nuclear factor erythroid-2-related factor 2 (NRF-2), but only Mabs R induced significantly the expression of the NRF2-regulated gene NQO1, denoting NRF2 activation with a significant difference between the S and the R variant (Fig 2A). Surprisingly, expression of the NRF2-regulated gene HMOX1 was not modulated upon Mabs infection (Fig 2A).

To confirm ROS production upon Mabs infection, Mabs-infected AOs were microscopically analysed after incubation with either MitoSOX or H2DCFDA to detect mitochondrial ROS and $H_2O_2$ production, respectively. As shown in Figs 2B–2D, both S- and R-Mabs infections enhanced the incorporation of MitoSOX and H2DCFDA, in agreement with the induction of ROS production. Interestingly, production of mitochondrial ROS and $H_2O_2$ was higher with the R variant, further confirming R-Mabs virulence compared to S-Mabs in AOs.

The contribution of cell protective antioxidant pathways during mycobacterial infection remains poorly understood [38,39]. We then determined the consequence of boosting antioxidant pathways for Mabs fitness. To assess the role of host-derived oxidative stress, Mabs-infected AOs were treated with the antioxidants resveratrol or the NRF2 agonist sulforaphane, which both significantly reduced by 70% S and R variant growth in AOs (Figs 2E and 2F, S3E and S3F), without affecting Mabs growth *in vitro* (S3D and S3G Fig). Interestingly, sulforaphane treatment also decreased the number of organoids containing Mabs R cords (Fig 2G).

Altogether, the results showed that the human airway organoids recapitulate features of the infection by Mabs variants. Moreover, independent of the immune system, airway epithelial cells mount an oxidative response upon Mabs infection, which contributes to Mabs growth and virulence.

## AOs recapitulate hallmarks of cystic fibrosis, and display enhanced oxidative stress

In order to evaluate CF lung context on Mabs infection, we next derived AOs from CF patients displaying Class I and II CFTR mutations (S1 Table). The Class II CF AO lines display the delF508 CFTR mutation bore by 80% of the CF patients [40]. Characterization of CF versus healthy AOs showed that CF AOs have impaired swelling in the forskolin assay (Fig 3A)

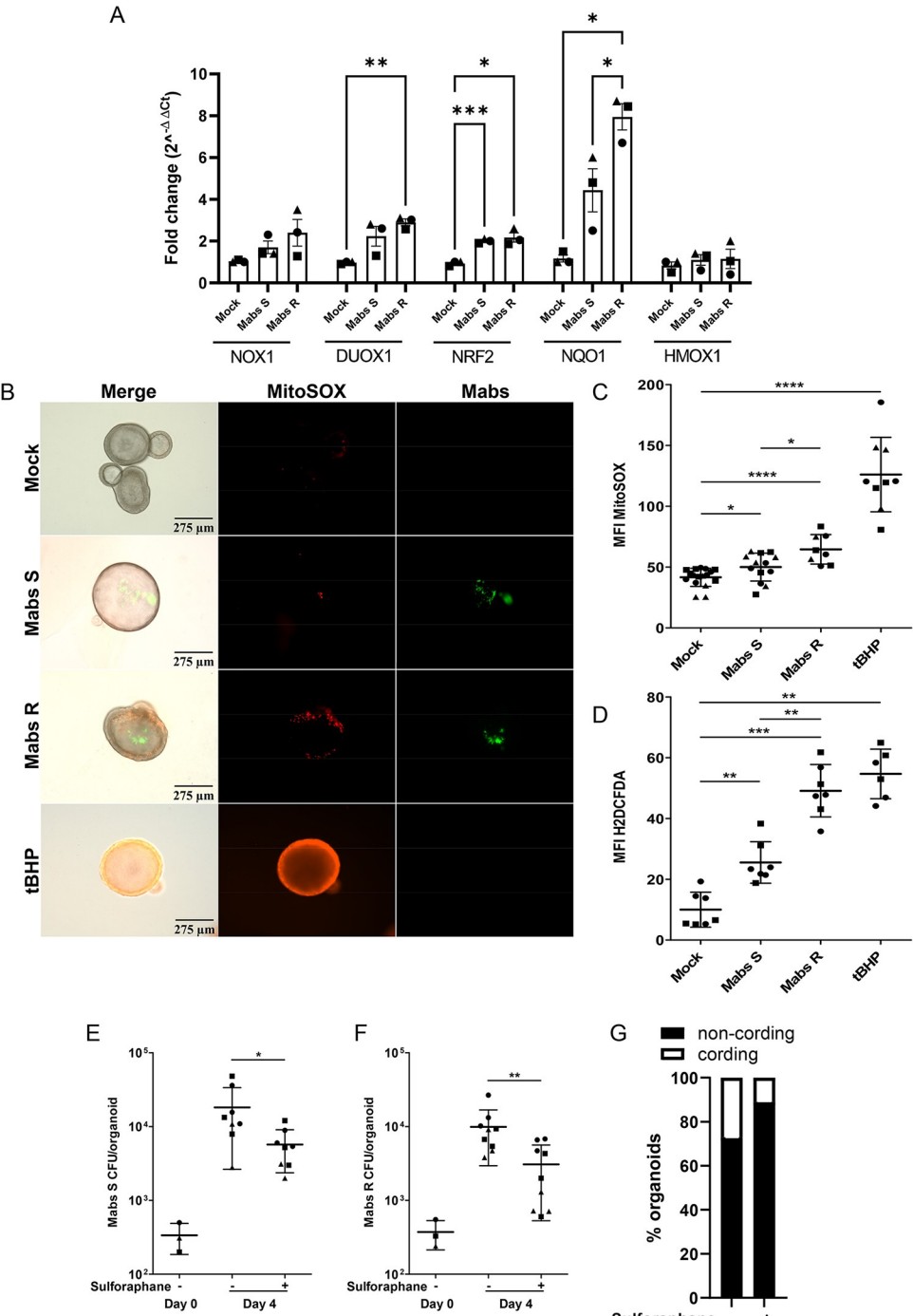

**Fig 2. S and R Mabs promote an oxidative environment in airway organoids.** (A) Expression patterns of ROS-related genes in mock-infected H-AOs or H-AOs infected with Mabs S or R for 4 days. Graph represents means ± SEM from three independent experiments, performed in triplicates. *P<0.05; **P<0.01; ***P<0.001 by unpaired T test. (B, C) Representative images (B) and MFI quantification (C) of mitochondrial ROS production (5µM MitoSOX) in mock-infected H-AOs (n = 17) or H-AO infected with Wasabi-labelled Mabs S (n = 13) or R (n = 8) for 3 days. (D) MFI quantification of $H_2O_2$ production (10µM H2DCFDA) in mock-infected AOs (n = 7) or AOs infected with tdTomato-labelled Mabs S (n = 7) or R (n = 7) for 3 days. (E, F) Bacterial load by CFU assay of H-AOs pre-treated with (+) or without (-) 10µM sulforaphane for 6hr before infection with Mabs S (E) (n+ = 8; n- = 8) or R (F) (n+ = 9; n- = 9) for 4 days. (G) Mean percentage of H-AOs exhibiting cords after 4 days of infection with Mabs R, in the presence (n+ = 9) or absence (n- = 8) of 10 µM sulforaphane treatment. Except otherwise stated, graphs represent means ± SD from at least two independent experiments, indicated by different symbols. Each dot represents one organoid. *P<0.05; **P<0.01; ***P<0.001, ****P<0.0001 by Mann-Whitney test.

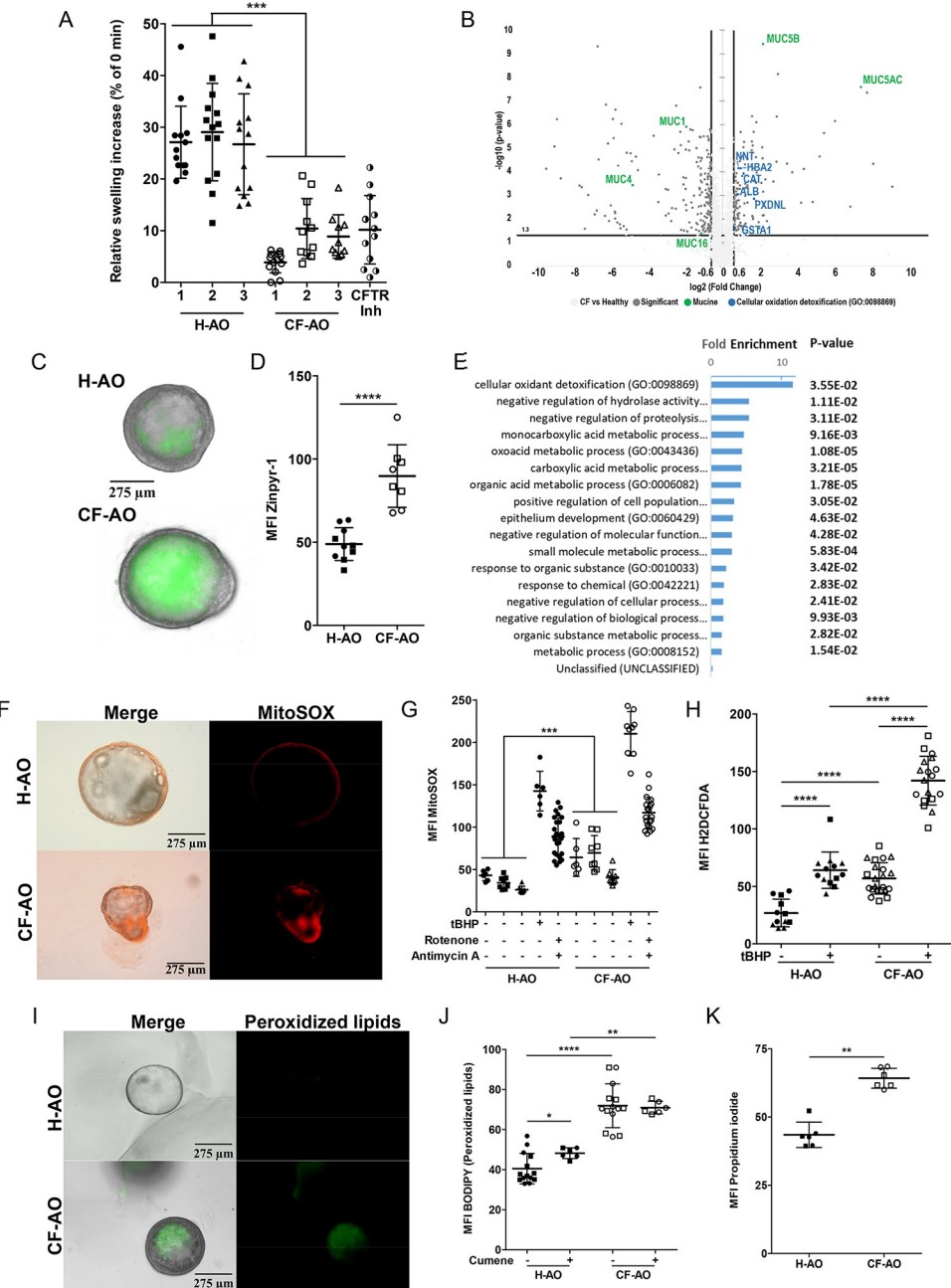

**Fig 3. Patient-derived airway organoids recapitulate cystic fibrosis-driven oxidative stress.** (A) Percentage of area increase of H-AOs (Donor 1 n = 13, Donor 2 n = 14, Donor 3 n = 13), CF-AOs (Donor 1 n = 13, Donor 2 n = 11, Donor 3 n = 10), and H-AOs pre-treated with CFTR inhibitors (25μM CFTRinh-172 and GlyH 101 for 4 days) (CFTR-Inh n = 13) after 2hr stimulation with 5μM forskolin. Data from two independent experiments. (B) The volcano plot showing the fold-change (x-axis) versus the significance (y-axis) of the proteins identified by LC–MS/MS in CF-AOs *vs* in H-AOs. The significance (non-adjusted p-value) and the fold-change are converted to −Log10(p-value) and Log2(fold-change), respectively. (C, D) Representative images (C) and MFI quantification (D) of mucus staining (10μM Zinpyr-1) in H-AOs (n = 10) and CF-AOs (n = 8). (E) Gene Ontology enrichment analysis showing the most enriched Biological Processes and their associated p-values (calculated using the Bonferroni correction for multiple testing) related to the list of up-regulated proteins in CF patients compared to healthy ones. (F, G) Representative images (F) and MFI quantification (G) of basal mitochondrial ROS production (5μM MitoSOX) in H-AOs (Donor 1 n = 6, Donor 2 n = 8, Donor 3 n = 8) and CF-AOs (Donor 1 n = 6, Donor 2 n = 8, Donor 3 n = 8). Data from two independent experiments. As positive controls for ROS production, two wells of healthy or CF organoids were treated for 1hr at 37˚C with 20mM tert-Butyl hydroperoxide (tBHP) (H-AOs n = 6; CF-AOs n = 9) or a mix of 5μM rotenone and 5μM antimycin A (H-AOs n = 26; CF-AOs n = 21). (H) MFI quantification of basal $H_2O_2$

production (10µM H2DCFDA) in H-AOs (-tBHP n = 12; +tBHP n = 13) and CF-AOs (-tBHP n = 22; +tBHP n = 18). (I, J) Representative images (I) and MFI quantification (J) of peroxidized lipids (2µM BODIPY) in H-AOs (n = 14) and CF-AOs (n = 14). As positive control for lipid peroxidation induction, healthy (n = 6) or CF (n = 6) AOs were treated with 800µM cumene hydroperoxide for 2hr at 37˚C. (K) MFI quantification of the basal plasma membrane permeabilization (50 µg ml$^{-1}$ propidium iodide incorporation) in H-AOs (n = 6) and CF-AOs (n = 6). Except otherwise noted, graphs represent means ± SD from at least two independent experiments indicated by different symbols. Each dot represents one organoid. *P<0.05; **P<0.01; ***P<0.001, ****P<0.0001 by Mann-Whitney test.

denoting CFTR channel malfunction, similar than pharmacological inhibition of the CFTR. They also displayed a thicker epithelium (S4A and S4B Fig). Mass spectrometry analysis of CF vs healthy AOs revealed enhanced abundance of mucins MUC5AC and MUC5B (Fig 3B) that was confirmed at the mRNA level (S4C Fig) and by exacerbated accumulation of mucus in the CF AO lumen (Figs 3C, 3D and S4D). A gene ontology enrichment study revealed tendency towards an upregulation of the cellular oxidant detoxification pathway in CF AOs (fold-change 11.6; p-value 3.55e-2; Fig 3E). To further evaluate the oxidative status in the CF context, healthy and CF AOs were stained with MitoSOX or H2DCFDA. The ROS production was enhanced in CF AOs compared to healthy ones (Fig 3F–3H), which was further exacerbated after treatment with the oxidative stress inducer tert-Butyl hydroperoxide (tBHP) [41], and rotenone and antimycin A, inhibitors of the mitochondrial electron transport, both treatment inducing higher oxidative stress in CF-AOs than in healthy ones (MFI MitoSOX 142.5+9.4 vs 210.2+8.7 (*P*<0.001) for tBHP, and 88.8+4.2 vs 117+3.9 (*P*<0.0001) for rotenone/antimycin A, in H-AOs compared to CF ones) (Figs 3G, 3H and S4E). Because high ROS levels induce lipid peroxidation leading to cell death [42–44], we examined and quantified these processes by microscopy using BODIPY (measuring lipid peroxidation) or propidium iodide (measuring cell death) respectively. We found that level of peroxidized lipids (Figs 3I, 3J and S4F) and cell death (Figs 3K and S4G) were higher in CF AOs than in healthy AOs. Interestingly, while the oxidizing agent cumene hydroperoxide enhanced lipid peroxidation in H-AOs, it had no additional effect in CF-AOs (Figs 3J and S4F).

Altogether, these results showed that organoids derived from CF lung tissue exhibited not only CFTR dysfunction and exacerbated mucus accumulation but also an increased oxidative stress, therefore representing a suitable *ex vivo* model to investigate how the lung CF context drives Mabs infection.

### *Mycobacterium abscessus* takes advantage of CF-driven oxidative stress to thrive

As oxidative stress is enhanced in CF-AOs, we hypothesised that the CF context could favour Mabs growth. To test this hypothesis, we microinjected healthy and CF-AOs with S- or R-Mabs variants and quantified Mabs proliferation. At 4 days post-infection, we observed an enhanced replication for both variants in CF-AOs compared to in H-AOs (Fig 4A and 4B), indicating that the CF environment favours Mabs fitness. Interestingly, we quantified a higher percentage of organoids containing Mabs R cords in CF AOs (Fig 4C and 4D), suggesting a higher virulence of Mabs R in the CF context. To support these results, we next used CFTR inhibitors. First, we showed that treatment of H-AOs with CFTR inhibitors enhanced oxidative stress and cell death, hallmarks of CF-AOs (S5B and S5C). While CFTR inhibitors by themselves did not modify Mabs growth *in vitro* (S5D Fig), they enhanced Mabs proliferation in AOs compared with untreated ones (S5E and S5F Fig), associated with enhanced cording for Mabs R (S5G and S5H Fig), thus confirming that alteration of CFTR function promoted Mabs growth.

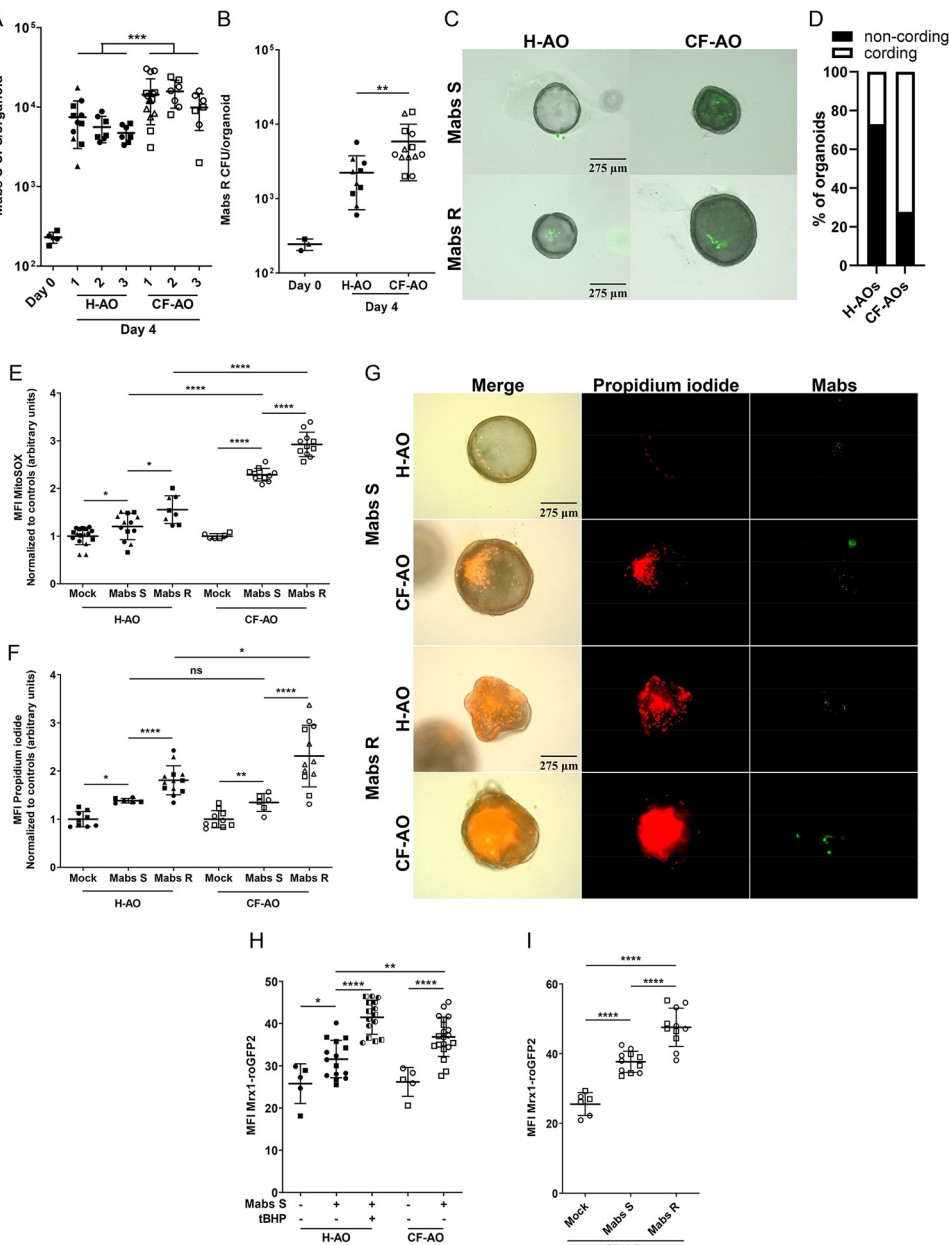

**Fig 4. Oxidative stress in cystic fibrosis benefits Mabs growth.** (A, B) Bacterial load by CFU assay of H-AOs and CF-AOs infected for 4 days with Mabs S (A) (healthy Donor 1 n = 11, Donor 2 n = 7, Donor 3 n = 7; cystic fibrosis Donor 1 n = 15, Donor 2 n = 7, Donor 3 n = 7) or R (B) (n healthy = 10; n cystic fibrosis = 13). (C) Representative images of Wasabi-labelled Mabs S or R 4 days-infected H-AOs and CF-AOs. (D) Mean percentage of H-AOs (n = 12) and CF-AOs (n = 7) exhibiting Mabs R cords after 4 days of infection. (E) MFI quantification of mitochondrial ROS production (5μM MitoSOX) in H-AOs and CF-AOs infected with Wasabi-expressing Mabs S (H-AOs n = 13; CF-AOs n = 11) or Mabs R (H-AOs n = 8; CF-AOs n = 11) for 3 days. Data are normalized to their respective Mock-infected H-AOs and CF-AOs (H-AOs n = 17; CF-AOs n = 6) control (F, G) MFI quantification (F) and representative images (G) of propidium iodide incorporation (50 μg ml$^{-1}$) in H-AOs and CF-AOs infected with Wasabi-expressing Mabs S (H-AOs n = 6; CF-AOs n = 6) or Mabs R (H-AOs n = 13; CF-AOs n = 12) for 4 days. Data are normalized to their respective Mock-infected H-AOs and CF-AOs (H-AOs n = 9; CF-AOs n = 10) control. (H) MFI quantification of GFP in Mock-infected H-AOs and CF-AOs (H-AOs n = 5; CF-AOs n = 5) or H-AOs and CF-AOs infected with Mrx1-roGFP2-expressing Mabs S (H-AO n = 15; CF-AO n = 20) for 4 days. As a positive control for ROS induction, H-AOs (n = 15) were exposed to 200μM tBHP for 1 hour prior infection, which was maintained all along the experiment. (I) MFI quantification of GFP in Mock-infected CF-AOs (n = 6) or CF-AOs infected with Mrx1-roGFP2-expressing Mabs S (n = 11) or Mabs R (n = 11) for 4 days. Except otherwise noted, graphs represent means ± SD from at least two independent experiments indicated by different symbols. Each dot represents one organoid. *P<0.05; **P<0.01; ***P<0.001; ****P<0.0001 by Mann-Whitney test.

Next, we showed that Mabs-infected CF-AOs exhibited higher oxidative stress and epithelial cell death than Mabs-infected H-AOs, indicating that CF-AOs exhibited higher susceptibility to Mabs infection, especially the R variant that induced higher cell death than the S one (Figs 4E–4G, S5I and S5J).

As Mabs infection and CF context are associated with host oxidative stress in airway organoids, we next wondered whether infection is associated with ROS production by the bacteria themselves. To address this question, we adapted the fluorescent redox biosensor Mrx1-roGFP2 system, initially developed in *M. tuberculosis*, to measure mycothiol redox potential in response to oxidative stress [45]. First, we evaluated ROS production *in vitro* upon oxidative agent exposure. Interestingly, at basal level, our results showed that the roGFP2ox/roGFP2red ratio is higher in Mabs S compared to Mabs R, indicative of a higher oxidized status in the S compared to the R variant (S6A Fig). By normalizing basal redox level to 1 in untreated conditions, we measured a slight increase in the roGFP2ox/roGFP2red ratio for Mabs S, which was significantly higher for Mabs R, indicating that the R variant displayed higher redox stress upon exposure to Diamide, tBHP or cumene hydroperoxide (CHP) (S6B Fig). Moreover, higher induction of ROS production by Mabs R in response to 5mM Diamide was associated with better bacteria survival, while exposure to 2mM tBHP or 250μM CHP resulted in bacterial killing (S6C Fig). Finally, Mrx1-roGFP2-expressing strains were microinjected in H- and CF-AOs, and GFP fluorescence intensity was microscopically quantified. As shown in Figs 4H and S6D, fluorescence of Mrx1-roGFP2 of Mabs S is enhanced in CF-AOs compared to healthy ones. Finally, Mrx1-roGFP2 fluorescence is higher in Mabs R than in Mabs S in CF context (Fig 4I).

Altogether, these results show that the CF context is associated with higher pathogenicity of Mabs revealed by enhanced bacterial growth and cording, bacterial and host oxidative stress and cell death.

## The NRF2-NQO1 axis supports a better control of Mabs growth

We next evaluated whether the antioxidant sulforaphane could inhibit Mabs growth. CF-AOs treated with sulforaphane exhibited mitigated oxidative environment (Figs 5A and S7A), and expressed higher level of NRF2-regulated NQO1 (Fig 5B). Activation of NRF2 resulted in reduced bacterial load (Fig 5C), % of organoids containing Mabs R cords (Fig 5D), and epithelial cell death (S7B and S7C), indicating that CF-driven oxidative stress stimulated Mabs growth and virulence, and host tissue damage. Interestingly, inhibition of NQO1 using dicoumarol abolished sulforaphane effect on Mabs growth and cording in CF context (Fig 5C and 5D), indicating that NQO1 might play a protective role during Mabs infection.

Finally, we evaluated the potential of sulforaphane combined with cefoxitin, a first line antibiotic to treat Mabs-infected patients [46]. First, we evaluated the influence of both treatment on ROS production upon Mabs S infection, and showed that sulforaphane, alone or in combination with cefoxitin, significantly inhibited host ROS production in CF-AOs upon Mabs S infection (Figs 5E and S7D). Moreover, while both cefoxitin and sulforaphane alone significantly inhibited Mabs growth in CF-AOs by 92% and 80% respectively, combination of both compounds was more efficient to reduce bacterial load (99% inhibition, Fig 5F). Of note, neither sulforaphane nor cefoxitin have any effect of the bacterial redox status (S7E Fig).

Altogether, these results show that pharmacological boost of antioxidant pathways, at least in part mediated by the NRF2-NQO1 axis, could be a complement strategy to current antibiotic therapies.

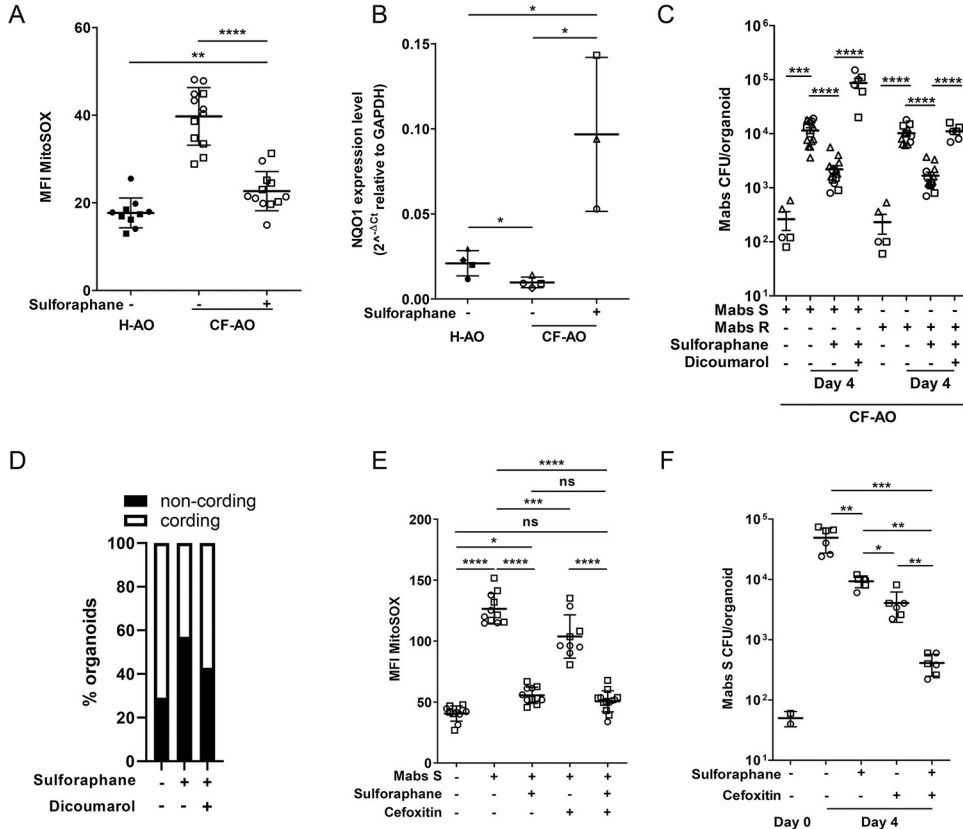

**Fig 5. The NRF2-NQO1 axis mitigates oxidative stress and Mabs growth in CF-AOs.** (A) MFI quantification of mitochondrial ROS production (5μM MitoSOX) in H-AOs (n = 10) and CF-AOs (n+ = 12; n- = 12) after 4 days of being treated with (+) or without (-) 10μM sulforaphane. (B) Expression of the NRF-2-regulated gene NQO1 in H-AOs and CF-AOs after 4 days of being treated with or without 10μM sulforaphane. Graph represents means from at least three pooled independent experiments, performed in triplicates. *P<0.05 by unpaired T test. (C) Bacterial load by CFU assay in CF-AOs pre-treated or not with 10μM sulforaphane for 6 hr before infection with Mabs S (treated n = 13; untreated n = 14) or Mabs R (treated n = 13; not-treated n = 12) for 4 days. When stated, the NQO1 inhibitor was added (10μM dicoumarol) (Mabs S n = 6; Mabs R n = 6) simultaneously with sulforaphane and maintained all along the experiment. (D) Mean percentage of CF-AOs untreated (n = 6) or treated with 10μM sulforaphane alone (n = 7) or in combination with 10μM dicoumarol (n = 7), exhibiting Mabs R cords after 4 days of infection. (E) MFI quantification of mitochondrial ROS production (5μM MitoSOX) in CF-AOs pre-treated or not with 10μM sulforaphane for 6 hr before infection with Wasabi-expressing Mabs S (treated n = 11; untreated n = 11) for 3 days. When stated, at day 2 post-infection, 20μg/ml of cefoxitin was added with (n = 13) or without (n = 9) 10μM sulforaphane. (F) Bacterial load by CFU assay of CF-AOs pre-treated with (n = 6) or without (n = 6) 10μM sulforaphane for 6 hr before infection with Mabs S for 4 days. When stated, at day 2 of the infection, 20μg/ml of cefoxitin was added with (n = 6) or without (n = 6) 10μM sulforaphane. Except otherwise stated, graphs represent means ± SD from at least two independent experiments indicated by different symbols. Each dot represents one organoid. *P<0.05; **P<0.01, ***P<0.001, ****P<0.0001 by Mann-Whitney test.

## Discussion

In this study, we used healthy and CF-patient derived airway organoids (AOs), as realistic and animal testing-free method, to assess Mabs pathogenicity in the context of natural CFTR functional alterations. Indeed, CF AOs carry the patient's own mutations and recapitulate key features of CF disease. We showed that both S- and R-morphotypes of *M. abscessus* proliferate and exhibit infection hallmarks in AOs, with the R variant displaying higher virulence in the CF context characterized by enhanced cording, host oxidative stress and cell death. We also demonstrated that boosting of antioxidant pathways might be a potential complement therapeutic strategy to current antibiotic treatment to better control Mabs infection in CF patients.

The organoid technology has demonstrated its usefulness in developing CFTR corrector/modulator therapies for CF patients [47–49]. Indeed, CF patient-derived organoids, bearing natural CFTR mutations, recapitulate *ex vivo* the spectrum of CFTR dysfunctions and CF disease severities [24,50]. Extended to the airway, CF patient-derived AOs have been shown to display epithelium hyperplasia, luminal mucus accumulation and abrogated response to forskolin-induced swelling, thus recapitulating CFTR dysfunction and consequences on the airway homeostasis [22]. Here, we derived AO lines from three CF patients, carrying class I and II mutations that reproduce the expected defective response to forskolin-induced swelling, epithelium hyperplasia and mucus accumulation. Moreover, these CF AOs display enhanced oxidative stress and lipid peroxidation, as previously measured in CF patients [36,37] or *in vitro* in CFTR mutated cell lines [51], as well as enhanced cell death, recapitulating CF-driven tissue damage. Therefore, our results demonstrate that CFTR dysfunction in epithelial cells is sufficient to cause an oxidative status imbalance in the airway epithelium, independent of immune cells and infection.

We and others have already applied the organoid technology to model host-pathogen interactions [22,26,30,31,52,53]. Here, we have reproduced Mabs infection hallmarks in AOs in healthy and CF contexts. Specifically, we show that both S- and R-Mabs replicate as extracellular bacteria in AOs, which was further enhanced in CF context, thus recapitulating the susceptibility of CF patients to NTM infection [54,55] and consistent with Mabs localization in the airway of CF patients [56]. We showed that Mabs S is surrounded by an extracellular substance resembling a biofilm. Visualizing bacterial biofilm still remains a challenge, especially in *in vivo* settings. The formation of biofilm during both acute and chronic infection plays a crucial role at protecting extracellular bacilli from immune response and antimicrobial agents, leading to treatment failure [57–59]. Moreover, it is now recognized that biofilms are highly diverse bacterial communities, which depend on the environmental conditions, hardly reflected by *in vitro* cultures in laboratories [60]. Therefore, the detection of biofilm in Mabs S-infected healthy and CF AOs opens new venues to further dissect how the human airway, in healthy but also pathological contexts, influences Mabs biofilm formation and consequences on infection, and for testing antibiofilm activity of novel pharmacological compounds [57,61,62].

We also showed that the R variant forms serpentine cords, structures described to be associated with mycobacteria virulence [6–8,11,35,63–67]. Interestingly, formation of serpentine cords by Mabs R in AOs was modulated by the host responses. Indeed, Mabs R cords were increased over time of infection, enhanced in the CF context, and decreased after antioxidant treatment, also correlating with reduced levels of host oxidative stress and cell death. How host pathways can influence mycobacterial cording and virulence has been extensively described in the zebrafish model infected with *M. marinum* or *M. abscessus*, showing that a fine-tuned regulation of the host inflammatory response is crucial to ensure a proper control of mycobacterial infection [15,63,68–70]. It is interesting to note that mycobacteria cords display their virulence trait both intracellularly by triggering phagosomal rupture and death of infected cells [66,71–73], but also extracellularly by escaping phagocytosis and thus contributing to bacterial persistence and tissue damage [32]. Therefore, AOs constitute a relevant model to decipher the extracellular lifestyle of Mabs in the bronchial tree where biofilm and cording constitute key structures of Mabs variant pathobiology [74,75].

ROS production is a part of host antimicrobial defence but requires a fine-tuned balance to prevent tissue damage. Indeed, the production of ROS is essential to control infection, as knock-down of NOX2 expressed in immune cells, or inhibition of DUOX2-mediated epithelial ROS production resulted in uncontrolled bacteria proliferation in zebrafish and mouse models [15,76,77]. Here, we showed in airway organoids, that Mabs R infection triggers ROS production, which was further enhanced in CF context as previously described in zebrafish [15]. This

enhanced ROS production is associated with enhanced expression of the epithelial cell ROS production gene DUOX1, concomitantly to the expression of the antioxidant NRF2-NQO1 pathway. Interestingly, boosting the NRF2-NQO1 axis was associated with a better control of Mabs growth in AOs indicating that oxidative stress favours Mabs fitness. Indeed, activation of NRF2 with sulforaphane in CF organoids resulted in better control of Mabs, abolished by NQO1 inhibition using dicoumarol. Only few studies investigated the role of NRF2 in myco-bacteria infection [78]. In Mtb-infected macrophages, it has been shown that activation of NRF2 is associated with enhanced bacterial load and cell survival [79], while inhibition of NRF2 or NQO1 ensured a better control of Mtb or BCG infection and host cell survival [80,81], suggesting that NRF2-NQO1 axis, by detoxifying ROS, might play a detrimental role during mycobacteria infection. By contrast, Zhou et al. showed that the activation of NRF2-NQO1 axis induced better control of Mtb [82]. Regarding Mabs infection, as we observed in airway organoids, studies showed that antioxidants resulted in better control of the infection in macrophages and zebrafish [83–85], while a contradictory one showed that knock-down of NRF2, by enhancing ROS level, is associated with a better control of Mabs [86]. Here we showed that cumulative oxidative stress due to both the CF context and Mabs infection might result in a permissive environment for extracellular bacteria growth and the establishment of chronic infection in the lung of CF patients. Because antioxidants did not inhibit Mabs growth and bacterial redox status *in vitro*, our results indicate that ROS produc-tion by epithelial cells is sufficient to generate a permissive environment for extracellular Mabs proliferation, which could play a major role in the establishment of Mabs colonization of CF patient airway. Therefore, even with low bacteria internalization by epithelial cells [31] and in the absence of immune cells, AOs recapitulate the contribution of the host NRF2-NQO1 axis on bacteria fitness in the airway, which can be boosted pharmacologically to improve tissue redox homeostasis and to foster infection control in combination with frontline antibiotic treatments, as we exemplified here with cefoxitin. Further investigations are now warranted to integrate ROS production by immune cells, essential to control the intracellular pool of bacte-ria [15]. Interestingly, treating R Mabs-infected zebrafishes with resveratrol improves fish sur-vival and reduces bacterial load [84], suggesting that enhancing the global defence against ROS by antioxidants would improves Mabs infection control *in vivo*. Nevertheless, it would be interesting to extend the application of airway organoid by integrating innate immune cells to advance our understanding of mucosal immune responses, as previously described for NSCLC lung cancer and the evaluation of anti-tumoral immunotherapies [87,88]. Using a mouse model of CF, CCR2-mediated lung accumulation of monocyte-derived macrophages has been shown to drive neutrophil pulmonary inflammation upon LPS challenge [89]. As macrophages constitute an intracellular replicative niche for Mabs and display microbicidal deficiency in CF [90,91], the organoid technology constitutes a powerful human system to reconstruct the cellu-lar framework of the human airway epithelial-immune niche. This will require in the future to access patient-matched lung biopsy and blood sample in order to co-culture epithelial and immune cells with matched CFTR mutations. Finally, combination of the CFTR modulators and correctors Elexacaftor-Tezacaftor-Ivacaftor, recently approved by FDA (Trikafta) and EMA (Kaftrio) is a game-changer for the treatment of CF. It is a highly effective therapy designed to correct CFTR folding and function for the treatment of CF patients carrying at least one CFTR delF508 mutation (70% of the patients) [92]. It is associated with clinically meaningful improvements in lung function and respiratory-related quality of life. Since its introduction, clinical observations highlighted that CF patients show a rapid reduction of infection-related hospital visits and antimicrobial use after starting the therapy [93,94], sug-gesting that by restoring CFTR function, Kaftrio may improve lung ion homeostasis and/or restructure microbial niche. Interestingly, Kaftrio therapy eradicated NTM infection in CF

patients [95]. Interestingly, CF is associated with a defective NRF2 expression, contributing to the excessive oxidative stress and lung tissue damage, whereas CFTR modulators rescue NRF2 function and therefore improve tissue oxidative status [96]. Therefore, CFTR-targeted therapies, by restoring CFTR function and reducing lung tissue redox status, might confer a better control of bacterial infection. But how Kaftrio influences CF-driven new or already established Mabs infection warrants further investigation. By responding to Kaftrio [97], CF AOs constitute the unique model available to further decipher, in a personalized way, the consequences of Kaftrio therapy on CF-driven respiratory infection.

In conclusion, we have established AOs as a pertinent model of both CF airway dysfunction and susceptibility to Mabs infection. Moreover, we have identified the cell protective NRF2-NQO1 axis as a potential therapeutic target to restore CF tissue redox homeostasis and improve the control of bacteria growth. Therefore, our work opens new venues for deciphering the CF-associated extracellular lifestyle of Mabs, the role of critical host redox pathways and identifying innovative therapeutic intervention, as recently exemplified [98].

## Materials and methods

Detailed protocols are provided in the supplementary information (S1 Text).

### Ethics statements

The collection of patient data and tissue for AOs generation was performed according to the guidelines of the European Network of Research Ethics Committees following European and national law. The accredited ethical committee from CHU Toulouse reviewed and approved the study in accordance with the Medical Research Involving Human Subjects Act. Human lung tissue was provided by the CHU of Toulouse under protocol agreement (CHU 19 244 C and Ref CNRS 205782). As the biological materials consist in surgical waste repurposed for research, all patients participating in this study consented to scientific use of their material by verbal non-opposition statement; patients can withdraw their statement at any time, leading to the prompt disposal of their tissue and any derived material.

### Airway organoid culture and maintenance

Healthy adjacent tissue from three donors with lung cancer (women, age 65–67), and biopsies of lung tissue from three cystic fibrosis patients (S1 Table) were used to derive organoids as previously described with minor changes [22,31]. To prevent risk of microbial contamination, airway organoid complete media was supplemented with 10 μg ml$^{-1}$ Normocure (InvivoGen) and 2.5 μg ml$^{-1}$ Fungin (InvivoGen) during the first 4 weeks of the cystic fibrosis airway organoid cultures.

### Bacteria culture and organoid infection

*Mycobacterium abscessus sensu stricto* strain CIP104536T (ATCC19977T) morphotype S and R were grown as previously described [32]. Before infection, AOs were pretreated or no with 10μM Sulforaphane (Selleck Chemicals, Houston, TX, USA) for 6hr. Bacteria were prepared for microinjection as described before [31,99]. Briefly, bacterial pellets were resuspended in PBS 1x and disaggregated using a 1ml syringe (Terumo) with a blunt needle (Bio-Rad, Hercules, CA, USA). Bacterial density was adjusted to $OD_{600}$ = 0.1–0.4. Initial bacterial load is evaluated by microinjecting the same amount of bacteria in 100μl of PBS 1x and colony forming unit assay. As shown in S5A Fig, this approach gives a representative number of bacteria with reproducible microinjection between H- and CF-AOs. Infected organoids were individually

collected, washed in PBS 1x and embedded into fresh matrix (Matrigel (Corning)). Infected organoids were cultured for 3–4 days if not otherwise stated. Sulforaphane was maintained throughout the experiment and refreshed every two days. When stated, either prior Mabs infection or the second day post infection, 10μM dicoumarol (Sigma-Aldrich) or 20μg/ml cefoxitin (Sigma-Aldrich) were added, with or without sulforaphane.

### Microscopy

Live imaging was performed to visualize, quantify and assess cell death with 50 μg ml$^{-1}$ Propidium Iodide (Thermo Scientific), ROS levels by 10μM H2DCFDA (Invitrogen) or 5μM Mito-SOX (Thermo Scientific), CFTR function by forskolin-induced organoid swelling assay (2hr with 5μM Forskolin (Sigma-Aldrich)) as described [100], mucus accumulation with 10μM Zinpyr-1 (Santa Cruz Biotechnology), and lipid peroxidation by 2μM BODIPY (Thermo Scientific) in non-infected and Mabs-infected organoids. Images were acquired under an EVOS M7000 Imaging System and analyzed post-acquisition with Fiji/ImageJ.

For Lightsheet imaging, fixed (overnight in 4% paraformaldehyde) airway organoids were stained with propidium iodide (6μg/ml in PBS) for 30 minutes at room temperature then rinsed in PBS. Organoids were then embedded in 1% low-melting agarose inside glass capillaries and imaged in PBS using a light-sheet fluorescence microscope (Zeiss Lightsheet Z.1). The 3D reconstructions were performed with Amira software (v2020.2).

For scanning electron microscopy (SEM) and transmission electron microscopy (TEM), organoids were fixed in 2% paraformaldehyde (EMS, Hatfield, PA, USA), 2.5% glutaraldehyde (EMS) and 0.1 M Sodium Cacodylate (EMS). Samples were embedding in Durcupan ACM resin (Sigma-Aldrich) then, semi-thin (300nm) serial sections were made using an UC7 ultramicrotome (Leica, Wetzlar, Germany) and collected on silicon wafers (Ted Pella, Redding, CA, USA). Sections were imaged on a Quanta FEG 250 SEM microscope in BSE mode. Ultrathin sections were also collected on copper grids formvar coated for TEM analysis on a JEOL 1200 EXE II microscope.

### Colony forming unit (CFU) assay

At the indicated time point, organoids infected with Mabs were individually harvested and lysed in 100μl of 10% Triton X-100 (Euromedex) in cell culture grade water (Corning). Serial dilutions (factor of 10) were done to yield $10^{-1}$ to $10^{-5}$ dilutions of the original lysate and then plated on LB Agar (Invitrogen) in 55mm dishes (Sarstedt). Colonies were counted after an incubation of 4 days at 37°C.

### RT-qPCR

Organoids were collected at day 4 post-infection or stimulation and processed as reported [31]. Briefly, total RNA was extracted from organoids (15 AOs per condition) using the RNeasy mini kit (Qiagen) and retrotranscribed (150ng) with the Verso cDNA Synthesis Kit (Thermo Scientific). mRNA expression was assessed with an ABI 7500 real-time PCR system (Applied Biosystems) and the SYBR Select Master Mix (Thermo Scientific). Relative quantification was determined using the 2^-ΔΔCt or 2^-ΔCt method, normalized to GAPDH. Primer sequences are provided in S2 Table.

### Statistics

Statistical analyses were performed using Prism 8 and 5 (GraphPad Software). Data were compared by Mann-Whitney or unpaired T test and results reported as mean with SD. Data statistically significant was represented by *$P<0.05$; **$P<0.01$; ***$P<0.001$ and **** $P<0.0001$.

## Supporting information

**S1 Text. Supplementary Information.**
(DOCX)

**S1 Fig.** (A) Kinetics of Mabs S and R growth in H-AO. Graph shows three pooled independent experiments. (B) Growth of *Mycobacterium abscessus subspecie abscessus* (Day 0 n = 3; Day 4 n = 6), *subspecie massiliense* (Day 0 n = 3; Day 4 n = 6), and *subspecie bolletii* (Day 0 n = 3; Day 4 n = 6) in H-AO. Graph shows means ± SEM from two independent experiments. Each dot represents one organoid. **$P < 0.01$ by Mann-Whitney test. (C) Kinetics of *in vitro* growth of *Mycobacterium abscessus subspecie abscessus*, *subspecie massiliense*, and *subspecie bolletii*. Graph represents means from one experiment performed in triplicates. (D-I) 3D light-sheet imaging of airway organoids infected with wasabi (green) Mabs S or R. H-AOs were fixed then stained with propidium iodide to visualize cell nuclei (red) before imaging using Zeiss Lightsheet 1 microscope. (D) XY planes at the indicated z positions of the 400 µm z-stack of an H-AO after infection with Mabs S shown in S1 Movie (10X objective). (E) 3D visualization using AMIRA software of the z-stack of AO after infection with Mabs S. (F) Corner cut from two different angles using AMIRA software through a volume rendering of the nuclei while keeping the Mabs S fluorescent signal. Scale bar: 50 µm. (G) XY planes at the indicated z positions of the 400 µm z-stack of a H-AO after infection with Mabs R shown in S2 Movie (10X objective). (H) 3D visualization using AMIRA software of the z-stack of AO after infection with Mabs R. (I) Corner cut from two different angles using AMIRA software through a volume rendering of the nuclei while keeping the Mabs R fluorescent signal. Scale bar: 50 µm.
(TIF)

**S2 Fig.** (A) Rows 1 and 2: Transmission electron micrographs of healthy AO infected with Mab-S. Bacteria were found dispersed in loose aggregates in the layer close to the luminal side of the lung epithelium (black asterisk) and excluded from the mucus (white asterisk). Bacteria were not in direct contact with each other and did not show any preferred orientation. An accumulation of fibril-granular material was observed around the aggregates. (B) Transmission electron micrographs of healthy AO infected with Mab-R. Bacteria were found exclusively in the mucus (white asterisk) organized in bundles showing individual cells oriented and in close apposition with each other inside the same bundle.
(TIF)

**S3 Fig.** (A-C) Expression pattern of inflammatory cytokines (A), antimicrobial peptides (B), and mucins (C) in mock-infected H-AO, or H-AO infected with Mabs S or R for 4 days. Graphs represent means ± SEM from at least three independent experiments, performed in triplicates. *$P < 0.05$; ***$P < 0.001$; ****$P < 0.0001$ by unpaired T test. (D) Kinetics of *in vitro* Mabs S and R growth in absence or presence of 10µM resveratrol. Graph represents means from one experiment performed in triplicates. ns = not significative by Mann-Whitney test. (E-F) Bacterial load by CFU assay of H-AO pre-treated with (+) or without (-) 10µM resveratrol for 1hr before infection with Mabs S (E) (n+ = 7; n- = 6) or R (F) (n+ = 8; n- = 6) for 4 days. Graphs represent means ± SD from at two independent experiments, indicated by different symbols. Each dot represents one organoid. *$P < 0.05$; **$P < 0.01$ by Mann-Whitney test. (G) Kinetics of *in vitro* Mabs S and R growth in absence or presence of 10µM sulforaphane. Graph represents means from one experiment performed in triplicates. ns = not significative by Mann-Whitney test.
(TIF)

**S4 Fig.** (A, B) Representative bright-field images (A) and quantification (B) of epithelium thickness in healthy AOs (H-AO n = 32) and cystic fibrosis AOs (CF-AO n = 24). Data from three independent wells per donor. (C) Basal expression of mucin genes in H-AO and CF-AO. Graph represents means from three pooled independent experiments, performed in triplicates. **P<0.01; **** P<0.0001; ns = not significant by unpaired T test. (D) Electron micrographs of H-AO and CF-AO revealing mucus accumulation in the lumen and longer cilia in the CF ones. (E) Representative images of mitochondrial ROS production (5μM MitoSOX) in H-AO and CF-AO after 1hr treatment with 20Mm tBHP or a mix of 5μM rotenone and 5μM antimycin A. (F) Representative images of peroxidized lipids (2μM BODIPY) in H-AO and CF-AO after 2hr treatment with 800μM cumene hydroperoxide. (G) Representative images of the basal propidium iodide incorporation (50 μg ml$^{-1}$) in H-AO and CF-AO.
(TIF)

**S5 Fig.** (A) Initial bacterial load evaluation by CFU assay of H-AO and CF-AO infected with Mabs S (H-AO n = 6; CF-AO n = 6) or Mabs R (H-AO n = 6; CF-AO n = 6). (B) MFI quantification of mitochondrial ROS production (5μM MitoSOX) in H-AO after 4 days of being treated with (+ n = 16) or without (- n = 13) 25μM CFTR inhibitors (CFTRinh-172 and GlyH 101). (C) MFI quantification of propidium iodide incorporation (50 μg ml$^{-1}$) in H-AO after 4 days of being treated with (+ n = 10) or without (- n = 8) 25μM CFTR inhibitors. (D) Kinetics of Mabs S and R growth in absence or presence of 25μM CFTR inhibitors. Data from one experiment performed in triplicates. (E, F) Bacterial load by CFU assay of H-AO pre-treated with (+) or without (-) 25μM CFTR inhibitors for 2 days before infection with Mabs S (E) (n+ = 6; n- = 6) or R (F) (n+ = 8; n- = 6) for 4 days. (G) Representative images of H-AO pre-treated or not with 25μM CFTR inhibitors for 2 days before infection with Mabs S or R for 4 days. (H) Mean percentage of H-AO untreated (n = 18) or treated (n = 30) with 25μM CFTR inhibitors exhibiting cords after 4 days of infection with Mabs R. (I) MFI quantification of propidium iodide incorporation (50 μg ml$^{-1}$) in Mock-infected H-AO and CF-AO (H-AO n = 9; CF-AO n = 10) or H-AO and CF-AO infected with Wasabi-labelled Mabs S (H-AO n = 6; CF-AO n = 6) or Mabs R (H-AO n = 13; CF-AO n = 12) for 4 days. (J) MFI quantification of mitochondrial ROS production (5μM MitoSOX) in Mock-infected CF-AO (CF-AO n = 6) or CF-AO infected with Wasabi-labelled Mabs S (CF-AO n = 11) or Mabs R (CF-AO n = 11) for 3 days. Except otherwise stated, graphs represent means ± SD from at least two independent experiments indicated by different symbols. Each dot represents one organoid. *P<0.05; **P<0.01; ****P<0.0001; ns = not significant by Mann-Whitney test.
(TIF)

**S6 Fig.** (A) Basal ratiometric sensor response of Mrx1-roGFP2-expressing Mabs S and R measured. * *P*< 0.05, paired t test of three independent experiments. (B) Mabs S (red) and R (green) expressing Mrx1-roGFP2 were either left untreated (Mock) or exposed to different concentrations of diamide, tert-Butyl hydroperoxide (tBHP) or cumene hydroperoxide (CHP), and the ratiometric sensor response (405/480 ratio) was measured after 2 h post-exposure. Data represent the Mean± SD of three independent experiments. For S to R comparison, Mock condition was normalized to 1. S to R comparison: * P<0.05, ** P<0.01, *** P<0.001, **** P< 0.0001; Mabs variant to their respective Mock control: # P<0.05, ## P< 0.01, ### P< 0.001. (C) Mabs S (red) and R (green) expressing Mrx1-roGFP2 were either left untreated (Mock) or exposed to different concentrations of diamide, tert-Butyl hydroperoxide (tBHP) or cumene hydroperoxide (CHP), and bacterial growth in untreated (Mock) or oxidative agent-treated conditions was measured by the McFarland technique measuring bacterial culture turbidity. (D) Representative images of H-AO and CF-AO treated or not with 200μM tBHP for 1

hour before infection with roGFP2-expressing Mabs S for 4 days.
(TIF)

**S7 Fig.** (A) Representative images of mitochondrial ROS production (5μM MitoSOX) in H-AO and CF-AO after 4 days of being treated or not with 10μM sulforaphane. (B, C) Representative images (B) and MFI quantification (C) of propidium iodide incorporation (50 μg ml⁻¹) in CF-AO pre-treated with (+) or without (-) 10μM sulforaphane for 6 hr before infection with Wasabi-labelled Mabs S (n+ = 6; n- = 6) for 4 days. (D) Representative images of MitoSOX staining in Mock- or Mabs-infected CF organoids treated or not with sulforaphane and cefoxitin, alone or in combination. (E) Mabs S expressing Mrx1-roGFP2 was either left untreated (Mock) or exposed to 5mM diamide, then left untreated or treated with 10μM sulforaphane and/or 20μg/ml cefoxitin, and the ratiometric sensor response (405/480 ratio) was measured after 2 h post-exposure. Data represent the Mean± SD of three independent experiments. * P<0.05, by Two-way ANOVA. Except otherwise stated, graphs represent means ± SD from at least two independent experiments, indicate them by different symbols. Each dot represents one organoid. **P<0.01; ***P<0.001 by Mann-Whitney test.
(TIF)

**S1 Movie. Representative 3D reconstruction of an S Mabs-infected human airway organoid showing green wasabi-expressing bacteria aggregates in the lumen.** Nuclei are revealed by staining with propidium iodide (red). Images were acquired on a Zeiss lightsheet 1 microscope, and combined for 3D visualization using the AMIRA software.
(MP4)

**S2 Movie. Representative 3D reconstruction of an R Mabs-infected human airway organoid showing green wasabi-expressing bacteria forming cords in the lumen.** Nuclei are revealed by staining with propidium iodide (red). Images were acquired on a Zeiss lightsheet 1 microscope, and combined for 3D visualization using the AMIRA software.
(MP4)

**S1 Table. Clinical data from CF patients who underwent lung transplantation and from who a lung biopsy was obtained to derive airway organoids.**
(DOCX)

**S2 Table. For each gene analyzed, list of forward and reverse primers and related bibliography.**
(DOCX)

## Acknowledgments

We thank Nicole Schieber (EMBL Heidelberg, Germany) for sharing with us the embedding protocol. We thank Veronique Richard and Franck Godiard from the "Microscopie Electronique et Analytique" service of the University of Montpellier for assistance in ultramicrotomy and TEM, respectively. We thank Bruno Payre from the "Centre de Microscopie Electronique pour la Biologie" of the University of Toulouse 3 for his assistance in SEM. The authors thank Sophie Manzi for technical assistance with the Mrx1-roGFP2 strains. This manuscript was edited at Life Science Editors.

## Author Contributions

**Conceptualization:** Stephen Adonai Leon-Icaza, Salimata Bagayoko, Romain Vergé, Nino Iakobachvili, Chloé Ferrand, Talip Aydogan, Célia Bernard, Angelique Sanchez Dafun,

Marlène Murris-Espin, Julien Mazières, Pierre Jean Bordignon, Pascale Bernes-Lasserre, Victoria Ramé, Jean-Michel Lagarde, Julien Marcoux, Marie-Pierre Bousquet, Christian Chalut, Hans Clevers, Peter J. Peters, Virginie Molle, Geanncarlo Lugo-Villarino, Kaymeuang Cam, Laurence Berry, Etienne Meunier, Céline Cougoule.

**Funding acquisition:** Peter J. Peters, Virginie Molle, Geanncarlo Lugo-Villarino, Laurence Berry, Etienne Meunier, Céline Cougoule.

**Investigation:** Stephen Adonai Leon-Icaza, Salimata Bagayoko, Nino Iakobachvili, Chloé Ferrand, Talip Aydogan, Célia Bernard, Pierre Jean Bordignon, Christian Chalut, Kaymeuang Cam, Céline Cougoule.

**Methodology:** Stephen Adonai Leon-Icaza, Salimata Bagayoko, Romain Vergé, Nino Iakobachvili, Chloé Ferrand, Talip Aydogan, Célia Bernard, Angelique Sanchez Dafun, Marlène Murris-Espin, Julien Mazières, Pierre Jean Bordignon, Serge Mazères, Pascale Bernes-Lasserre, Victoria Ramé, Jean-Michel Lagarde, Julien Marcoux, Marie-Pierre Bousquet, Christian Chalut, Hans Clevers, Peter J. Peters, Virginie Molle, Geanncarlo Lugo-Villarino, Kaymeuang Cam, Laurence Berry, Etienne Meunier, Céline Cougoule.

**Resources:** Marlène Murris-Espin, Julien Mazières, Jean-Michel Lagarde, Christophe Guilhot, Hans Clevers, Peter J. Peters, Virginie Molle, Laurence Berry, Etienne Meunier, Céline Cougoule.

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
