## [Decision Letter · Decision Letter 0]

23 Jan 2023

Dear Dr COUGOULE,

Thank you very much for submitting your manuscript "Druggable redox pathways against M. abscessus in cystic fibrosis patient-derived airway organoids" for consideration at PLOS Pathogens. As with all papers reviewed by the journal, your manuscript was reviewed by members of the editorial board and by several independent reviewers. In light of the reviews (below this email), we would like to invite the resubmission of a significantly-revised version that takes into account the reviewers' comments.

The authors need to ensure that differences in bacterial burden at the time of measurement of experimental parameters is not skewing data interpretation. The authors focused on the oxidative environment without necessarily providing data to measure the oxidative response and do explore the mechanistic basis for the differential virulence. It is important to address the concern regarding microinjection efficiency of healthy versus cystic fibrosis organoids.

We cannot make any decision about publication until we have seen the revised manuscript and your response to the reviewers' comments. Your revised manuscript is also likely to be sent to reviewers for further evaluation.

Sincerely,

Helena Ingrid Boshoff

Academic Editor

PLOS Pathogens

Marcel Behr

Section Editor

PLOS Pathogens

Kasturi Haldar

Editor-in-Chief

PLOS Pathogens

orcid.org/0000-0001-5065-158X

Michael Malim

Editor-in-Chief

PLOS Pathogens

orcid.org/0000-0002-7699-2064

The authors need to ensure that differences in bacterial burden at the time of measurement of experimental parameters is not skewing data interpretation. The authors focused on the oxidative environment without necessarily providing data to measure the oxidative response and do explore the mechanistic basis for the differential virulence. It is important to address the concern regarding microinjection efficiency of healthy versus cystic fibrosis organoids.

Reviewer's Responses to Questions

**Part I - Summary**

Reviewer #1: Leon-Icaza et al. develop and exploit human airways organoids as a model system to characterize host-pathogen interaction with an emphasis on the emerging pathogen Mycobacterium abscessus. On the one hand the authors investigate two well described morphotypes of M. abscessus, namely a smooth morphotype and the rough morphotype. The smooth morphotype is covered by a layer of glycopeptido lipids, while these surface molecules are missing/mislocated in the rough morphotype. Upon infection of airway organoids bacteria of both morphotypes multiply (over 12 days) and can trigger an oxidative stress response in the host which helps to control intracellular bacteria but also negatively affects viability of host cells. In the organoids, both morphotypes grow extra-cellularly but form either biofilm like-structure (smooth) or cords (rough), respectively. The R morphotype induces higher levels of oxidative stress. Organoids were derived from people suffering from Cystic Fibrosis and people with unrelated disease. Upon infection of CF-derived organoids a higher oxidative stress response is induced. A drug-mediated suppression of the oxidative stress response inhibits multiplication of the bacteria and favors the antibacterial effects of the b-lactam antibiotic cefoxitin, a first-line drug for the treatment of pulmonary M. abscessus infections.

The investigations demonstrate that the airway organoids model can mimic relevant in vivo features of a human disease (Cytic Fibrosis) and host-pathogen interaction. It provides the possibility to explore new antibacterial and host-directed therapies for an infection posed by a serious and hard-to-treat pathogen. The model may be run at a “medium through-put” size and significantly contribute to the 3R criteria in order to reduce animal experimentation.

In summary, the investigations demonstrate the model can recapitulate important features of host-pathogen interaction and may provide new insights into virulence mechanisms and hint to new therapeutic approaches.

Criticism: The entire investigation is conducted with one (set) of strains, namely M. abscessus type strain. It would be recommendable to extend the investigation to members of the other subspecies and to strains which show increased/decreased susceptibility to the drug(s) used for treatment.

Reviewer #2: In this study the authors establish airway organoids as a model to investigate the interactions between M. abscessus and the host epithelium. They generate organoids from “healthy” tissue as well as from cystic fibrosis patients and compare several parameters during infection with two distinct isolates of M. abscessus (a rough and a smooth one). This study is of interest for different reasons, including the description of the airway organoid model and its application during Mycobacterial infection, and the importance of non-tuberculous mycobacteria in the context of cystic fibrosis. Using this system the authors propose antioxidants as a potential host-directed strategy to improve M. abscessus infection control. Overall, the study is well designed and presented. Some aspects could be improved as pointed out below, mainly to advance the study beyond its descriptive nature.

Reviewer #3: The manuscript by Leon-Icaza et al., exploited previously reported human airway organoids (H-OEs) to evaluate Mabs fitness, host redox responses to infection, and efficacy of antioxidants. They further examined OEs from cystic fibrosis (CF) patients and showed the importance of antioxidants and CFTR inhibitors in controlling ROS and bacterial fitness. Finally, the authors proposed a combination of antioxidant(s) with cefoxitin as a potential host-directed strategy to improve Mabs infection control. Overall, the manuscript is interesting from a technology point of view and infection control. However, several conclusions need supportive evidence. More importantly, mechanistic insights on how host redox mechanisms modulate Mab fitness are crucial.

**Part II – Major Issues: Key Experiments Required for Acceptance**

Reviewer #1: -

Reviewer #2: In Figure 1 the authors infect (microinject) airway organoids with two isolates of M. abscessus and demonstrate that the bacteria grows within the organoids over time. They chose to perform the experiments on day 4 post-infection. Looking at the CFU dynamics (Figure 1A), there is app a 1 log difference in bacterial burden between the two M. abscessus isolates used on day 4. Would this difference in bacterial burden impact the remaining analysis? Also related to this first set of data, the authors describe several features of the bacterial distribution and interaction with the organoid cells, showing representative images. Quantifying these features would strengthen the data.

The authors then show that M. abscessus promotes a oxidative environment in the airway organoids. The rational for investigating specifically the oxidative environment, without considering other cellular responses was not clear. Is the oxidative response the most predominant one upon infection? Furthermore, some differences are shown with respect to the two M. abscessus isolates, but the mechanisms possibly underlying these differences are not discussed- they are justified based on the differential virulence of the M. abscessus isolate. Would the observed differences be for example linked to the differential structure of the smooth vs rough bacteria in the organoid, or their localization relatively to the epithelial cells? Providing some insights on these questions would advance the study beyond its descriptive nature.

The airway organoids derived from cystic fibrosis patients recapitulated important features of the disease, namely the epithelial dysfunctions. The authors follow their study by comparing the M. abscessus infection in healthy vs cystic fibrosis organoids. It is not clear whether the microinjection efficiency is similar in both organoid types, as the CFU shown for day 0 relates only to the healthy data. Given the somehow different basal structure/characteristics of the healthy vs cystic fibrosis organoids, it would be interesting to address this point. Related to this, the propidium iodide incorporation in mock conditions is already different between the two types of organoids (Figure 4D,E), which makes it difficult to assess whereas the differences seen upon infection reflects “higher susceptibility of cystic fibrosis organoids to Mabs infection”. It may be worth to quantify the fold increase in cell death relatively to mock conditions.

Reviewer #3: It needs to be clarified how an increase in oxidative stress promotes bacterial fitness. In general, higher ROS is associated with better control of mycobacterial infection. R-form is also better equipped to induce ROS than S-form. Both of these observations need in-depth characterization. Is Mabs more tolerant to oxidative stress than Mtb? Comparative survival of Mab vs Mtb in response to oxidants in vitro is important. The data need to be further supported by measuring the cytoplasmic redox potential of Mabs vs Mtb (or any other slow-growing mycobacteria that are known to infect macrophages). Similarly, experiments to show the cytoplasmic redox state of S-type vs R-type Mabs in vitro and inside H-OEs from healthy and CF patients are crucial to confirm improved bacterial fitness in response to higher oxidative stress.

Synergy experiments between antioxidants and cefoxitin can be strengthened by examining if potentiation is only in the context of OGs. Measuring host ROS and bacterial redox state in response to antioxidants and cefoxitin combination compared to antioxidant/ cefoxitin alone further bolsters the findings.

Lastly, why does the R-type induces higher oxidative stress than the S-type?

**Part III – Minor Issues: Editorial and Data Presentation Modifications**

Reviewer #1: Criticism: The entire investigation is conducted with one (set) of strains, namely M. abscessus type strain. It would be recommendable to extend the investigation to members of the other subspecies and to strains which show increased/decreased susceptibility to the drug(s) used for treatment.

Reviewer #2: Given the novelty of the model, it would be important for a general audience to provide more details in the methods section. For example, the section: “RT-qPCR and Colony forming unit assay. Organoids were collected at day 4 post-infection or stimulation and processed as reported (31).” does not provide any relevant information as to what and how has been done.

Despite the promising data shown in this study, and the potential of using airway organoids, the impact of immune cells in the system deserves a deeper discussion. It is true that the authors show a series of interesting findings even in the absence of immune cells, but their impact to the data needs further addressing.

Reviewer #3: Other comments.

Fig 1 A: Why are there differences at day 0 CFUs between S and R-types? Are there differences in internalization or initial establishment of infection?

Using the term latency to depict the lag shown by bacteria for the initial 2 days is inappropriate.

Fig. 1B/C/D” The images show clear bacteria. I am not convinced about the aggregate/biofilm nature of the S type or the serpentine nature of cords. Generally, cording improves with time and as bacteria lyse host cells and release in the outside milieu (extracellular). Necessary to show time-dependent changes in aggregate or cording behaviors? Quantification of cording needed (Cording/organoid)?

Fig. 1E/F shows the higher killing of epithelial cells, which cannot be equated to hypervirulence- better to avoid using hypervirulence to describe this phenomenon.

Fig. 2A: Induction of ROS-producing and antioxidants genes higher in R-type infected OEs than S-type? Please clarify this observation in the results section.

Fig. 2B/D: Why were these experiments done after three days of infection, whereas other experiments at four days? tPBH is not a good choice for assessing mitoROS; better to use antimycin D or rotenone?

Fig 2E/H: What is the effect of antioxidants on the cording? This would be important to show if the reduced bacterial growth is due to direct interference with phase switching.

Why CF AOs images look different in Fig 3C vs 3F? In 3C, the size of CF-AOs looks bulkier, which is not the case with 3C.

Fig 3I-J needs a positive control.

What is the effect of CFTR inhibitors on ROS and cytotoxicity upon infection?

Fig. 4H: Survival of Mabs is reduced upon antioxidant and antibiotic combination (not inhibition?).

Discussion: Cording also depends on other host factors (fh111, LTH4, TNf-Alpha)? These studies should be included in the discussion. Overall, the discussion could be more descriptive and more informative.

PLOS authors have the option to publish the peer review history of their article (what does this mean?). If published, this will include your full peer review and any attached files.

Reviewer #1: No

Reviewer #2: No

Reviewer #3: No
---

## [Editor Report · Decision Letter 1]

13 Jul 2023

Dear Dr COUGOULE,

We are pleased to inform you that your manuscript 'Druggable redox pathways against Mycobacterium abscessus in cystic fibrosis patient-derived airway organoids' has been provisionally accepted for publication in PLOS Pathogens.

Best regards,

Helena Ingrid Boshoff

Academic Editor

PLOS Pathogens

Marcel Behr

Section Editor

PLOS Pathogens

Kasturi Haldar

Editor-in-Chief

PLOS Pathogens

orcid.org/0000-0001-5065-158X

Michael Malim

Editor-in-Chief

PLOS Pathogens

orcid.org/0000-0002-7699-2064

The authors have sufficiently addressed the reviewers' comments.
---

## [Editor Report · Acceptance letter]

31 Jul 2023

Dear Dr COUGOULE,

We are delighted to inform you that your manuscript, "Druggable redox pathways against Mycobacterium abscessus in cystic fibrosis patient-derived airway organoids," has been formally accepted for publication in PLOS Pathogens.

Best regards,

Kasturi Haldar

Editor-in-Chief

PLOS Pathogens

orcid.org/0000-0001-5065-158X

Michael Malim

Editor-in-Chief

PLOS Pathogens

orcid.org/0000-0002-7699-2064